# Clinical Characterization of Pediatric Erythromelalgia: A Single-Center Case Series

**DOI:** 10.3390/children10081282

**Published:** 2023-07-26

**Authors:** Jenny Sun, Don Daniel Ocay, Meghan Halpin, Kimberly Lobo, Dafni F. T. Frohman, Carolina Donado, Catherine A. Brownstein, Casie A. Genetti, Anna Madden, Charles B. Berde

**Affiliations:** 1Department of Anesthesiology, Critical Care, and Pain Medicine, Boston Children’s Hospital, Boston, MA 02115, USA; 2Department of Anaesthesia, Harvard Medical School, Boston, MA 02115, USA; 3Manton Center for Orphan Disease Research, Division of Genetics and Genomics, Boston Children’s Hospital, Boston, MA 02115, USA

**Keywords:** erythromelalgia, sodium channelopathy, small-fiber neuropathy

## Abstract

Erythromelalgia is a descriptive term for severe burning pain and erythema in the distal extremities relieved by cold and exacerbated by heat. Pediatric case series to date are relatively small. We extracted and analyzed medical record data for 42 pediatric patients to describe clinical characteristics, associated conditions, and responses to treatments. Informed consent was obtained according to an IRB-approved protocol that included gene discovery. Three patients had confirmed Nav1.7 sodium channelopathies, with six additional patients under investigation with novel gene candidates. There was a female predominance (2.5:1), and the median onset age was 12 years (IQR = 3–14). Patients saw a median of three specialists (IQR = 2–3) for a diagnosis. The majority (90%) reported bilateral symptoms. Cooling methods usually provided partial relief, while heat and exercise exacerbated pain. No medication appeared to be consistently effective; commonly prescribed medications included sodium channel blockers (*n* = 37), topical analgesics (*n* = 26), gabapentin (*n* = 22), and aspirin (*n* = 15). Based on the currently published literature, we believe this cohort is the largest pediatric study of erythromelalgia to date. Many findings are consistent with those of previously published case series. Work is in progress to establish a prospective cohort and multi-center registry.

## 1. Introduction

Erythromelalgia is a descriptive term for a rare, chronic pain disorder involving an intense burning sensation, warmth, and erythema, usually involving the distal extremities [1,2]. Pain most commonly occurs bilaterally and symmetrically, with feet more often affected than hands, and less frequently in the face and ears [1]. Symptoms are worsened by ambient or local heat exposure and generally partially relieved by the application of local cooling, such as immersing extremities in ice water [1,3].

Erythromelalgia has been categorized in subtypes: primary, which may be sporadic or inherited, and secondary, when caused by other systemic diseases. However, in accordance with Arthur et al. [3], in this manuscript, patients with a confirmed genetic variant are determined as “inherited erythromelalgia,” while those without a confirmed genetic variant are termed “symptomatic” erythromelalgia. Challenges and controversies regarding disease classification and diagnosis will be summarized in the Discussion. Inherited erythromelalgia was initially linked to autosomal dominant gain-of-function mutations in the *SCN9A* gene, which encodes the Nav1.7 voltage-gated sodium channel (VGSC) [4]. Genetic variants with a similar clinical presentation have also been identified in other VGSCs, including *SCN10A* and *SCN11A* [5], as well as in genes associated with other ion channels (e.g., *TRPA1*) and with platelet–endothelial interactions (e.g., *JAK2*) [6,7]. In several previous case series, gene variants were identified in only a small subset of patients [4,8], though this percentage may change in the future with ongoing advances in gene sequencing and analyses.

The pathophysiology of inherited erythromelalgia associated with *SCN9A* variants has received extensive study using sodium channel electrophysiology in vitro [9,10,11]. The clinical observation of the exacerbation of pain by heat and relief by cold in vivo fits well with the observation of the altered temperature dependence of ionic currents in reconstituted channels in vitro.

Secondary erythromelalgia may be associated with a range of diseases, including Fabry disease, vasculitis, sarcoidosis, Sjögren’s syndrome, amyloidosis, and small-fiber neuropathy [12,13,14,15,16]. For these diseases, as well as for most other cases of non-inherited/symptomatic erythromelalgia, the detailed molecular mechanisms that give rise to burning pain, triggering with heat, and partial/temporary relief by cold are largely unknown.

There is limited information on the incidence or prevalence of erythromelalgia in different populations. A population-based study in Minnesota estimated the overall incidence to be 1.3 per 100,000 people per year [17]. Most case series report a female predominance, with a female to male ratio of about 2:1 [1,14,18].

A clinical diagnosis derives largely from the history and physical examination, often aided by pictures of affected areas taken during pain episodes. There is limited information to guide clinicians for how best to use clinical and epidemiologic variables to prioritize laboratory testing for systemic diseases that could present as symptomatic erythromelalgia [2,5]. Laboratory testing may in some cases lead to the diagnosis of treatable underlying conditions [12,13].

For a small subset of patients with inherited erythromelalgia, a genetic diagnosis may inform the selection of treatments [7,19,20]. For most other patients, pharmacologic treatment is largely based on custom and extrapolation, rather than on clinical trials or on underlying mechanisms. There is variability in the effectiveness of any single available therapy, ultimately leading to empirical trials of multiple medications in most patients [2]. Analgesics may be combined with non-pharmacological methods of preventing or minimizing symptoms, such as managing pain triggers [3]. Commonly prescribed medications for treating erythromelalgia include salicylates, sodium channel blockers, antidepressants, anticonvulsants, and topical analgesics [2,3,4].

A majority of published case series involved adults [1,21]. Pediatric case reports and case series have involved relatively small samples [4]. Arthur and colleagues reported on a systematic review of 25 pediatric cases with *SCN9A* mutations (among 28 case reports), as well as a more detailed analysis of 13 patients seen at Great Ormond Street Hospital for Children [3]. The largest pediatric sample of erythromelalgia published to date included 32 cases from the Mayo Clinic seen between 1970 and 2007 [4].

In order to provide an additional description of the spectrum of pediatric erythromelalgia, we conducted a retrospective medical record review of patients enrolled through a research program at Boston Children’s Hospital, to describe the demographic features, clinical course, treatments applied, results of gene sequencing, and other laboratory studies. We hypothesized that a majority of patients (1) would not have identifiable genetic causes, (2) would not have multiple family members affected, (3) would not have confirmed inflammatory or metabolic conditions, and (4) would be female.

## 2. Materials and Methods

### 2.1. Study Design

Under Institutional Review Board approval, we performed a retrospective cross-sectional analysis of patients with a presentation of erythromelalgia who already consented to and were enrolled in an existing Gene Discovery and repository protocol (IRB#: 10-02-0053) at the Manton Center for Orphan Disease Research at Boston Children’s Hospital (BCH).

The selection criteria included patients diagnosed with erythromelalgia based on their health records and reported symptoms of erythema, warmth, and burning pain improved by cold and worsened by heat. All available data from multiple clinic visits, laboratory reports, and diagnostic study reports were extracted by the first author (J.S.) from either BCH electronic health records (EHR) or from outside records and then consolidated into an Excel spreadsheet. In cases where there was uncertainty regarding clinical characteristics, the records were reviewed by the senior author (C.B.B.) to confirm their inclusion in this cohort. Privacy and confidentiality were maintained in accordance with HIPAA and BCH requirements.

### 2.2. Data Collection

Demographics included sex, age of symptom onset, age of diagnosis, and current age. We recorded the range of specialists seen, coexisting diagnoses, pain locations, characteristics, and responses to exacerbating and ameliorating stimuli. Mean least, average, and worst pain intensity scores, on a numeric rating scale (NRS) ranging from 0 (no pain) to 10 (worst pain imaginable), were calculated for each patient if scores were available for multiple clinic visits. Laboratory test data were recorded as available for complete blood count (CBC), C-reactive protein (CRP), erythrocyte sedimentation rate (ESR), rheumatoid factor (RF), thyroxine (T4), thyroid-stimulating hormone (TSH), Vitamin D, Vitamin B12, lead, Fabry and celiac testing, skin biopsies for neurite counting, and pertinent electrodiagnostic and imaging studies. For nutritional and endocrine tests, we documented all-time lowest and highest values. We recorded data on the prescribing of topical and systemic medications addressing the symptoms of erythromelalgia, and when available, impression of benefits and side effects as recorded in clinic notes. We documented the family history of erythromelalgia and other chronic, inflammatory, vascular, or neuropathic conditions.

### 2.3. Statistical Analysis

This was a convenience sample of patients enrolled in the Manton protocol up to 5 December 2022, and no formal sample size estimation was performed. Exploratory statistical analyses (Mann–Whitney U Test or Wilcoxon Rank Sum Test) were conducted in R studio [22] to compare the age of onset, average pain score, and worst pain score between patients with (SCN9A+) and without (SCN9A−) a confirmed *SCN9A* gene variant. Descriptive analyses were reported as the median and interquartile range (IQR), unless otherwise stated.

## 3. Results

### 3.1. Demographics, Onset, Comorbidities, and Family History

Forty-two patients were included in this case series. Table 1 shows the sample’s demographics and other existing comorbidities. The median age of onset was 12 years (IQR = 3–14) and the median age of diagnosis was 13 years (IQR = 7–15). The median time between the age of onset and diagnosis was 12 months (*n* = 39, IQR = 0–42). Fifteen (35.7%) patients reported some type of inciting event for their onset of erythromelalgia, shown in Table 1.

Six (14.3%) patients reported another family member with erythromelalgia and twenty-two (53.4%) patients had at least one family member with some type of chronic, inflammatory, vascular, or neuropathic condition. The most frequently reported conditions in family history were rheumatoid arthritis (*n* = 4), Raynaud’s (*n* = 4), irritable bowel syndrome (*n* = 4), osteoarthritis (*n* = 3), fibromyalgia (*n* = 3), migraines (*n* = 3), hand/foot sensory symptoms (*n* = 3), psoriasis (*n* = 2), and endometriosis (*n* = 2).

### 3.2. Clinical History

Patients were seen by a median of three specialists (IQR = 2–3), shown in Table 2.

### 3.3. Pain Characteristics

Thirty-seven (88.1%) patients reported bilateral hands and feet as affected extremities, while one (2.4%) patient reported pain only in bilateral feet without the involvement of hands. Twenty-four (57.1%) patients reported other affected body parts besides hands and feet (Table 3). Of these, two did not report having affected bilateral hands or feet. Aside from pain, 14 (33.3%) patients further reported swelling in hands and feet, while 3 (7.1%) reported only feet swelling without hand involvement. In total, 25 (59.5%) patients reported fatigue/sleep disturbance and 2 (4.8%) reported mental clouding.

Thirty-six (85.7%) patients reported pain quality narratively (Table 3). Of these, 5 patients reported only “burning,” while 23 experienced a combination of burning along with other pain-quality types.

Thirty-eight (90.5%) patients reported a coloration of their affected body areas, with red (*n* = 36) and purple (*n* = 7) being the most commonly reported. Out of the 36 patients who reported a red coloration of their affected extremities, 27 reported only red while the remaining 9 had a combination of red and other colors (Table 3).

Twenty-eight patients reported at least one pain score in their EHRs for least, average, and worst pain, summarized in Figure 1.

### 3.4. Modifying Factors

Modifying factors were classified into those that improved symptoms and those that exacerbated symptoms. In total, 23 (54.8%) patients reported at least one type of modifying factor that improved their symptoms while 32 (76.2%) patients reported at least one type of modifying factor that exacerbated their symptoms (Figure 2). “Temperature changes” were reported as both a modifying factor that improved and exacerbated symptoms. However, in the context of improving symptoms, patients would describe “temperature changes” as moving from extreme hot or cold temperatures to a comfortable ambient temperature. For exacerbating factors, the term “temperature changes” encompassed moving from an ambient temperature to cold or warm weather extremes, and humidity.

### 3.5. Laboratory Tests

Laboratory tests were recorded for 38 (90.5%) patients, with CRP (*n* = 29) and ESR (*n* = 28) as the two most common laboratory tests (Figure 3). Figure 3 also depicts the number of patients with an abnormal result for each type of laboratory test.

Four patients underwent skin biopsies. Three were interpreted as consistent with small-fiber neuropathy. Two of these three patients with positive skin biopsies also had genetically confirmed *SCN9A* variants.

Ten patients underwent electromyography and nerve conduction studies (EMG/NCS). Two of the ten patients who underwent EMG/NCS had findings interpreted as abnormal. One patient presented initially with classical erythromelalgia, with severe burning pain in all extremities and a near-continuous use of ice. Over the next 3 weeks, she developed distal weakness and hyporeflexia in all extremities. EMG/NCS showed mixed axonal and demyelinating polyneuropathy. She was treated for a presumptive auto-inflammatory process with steroids and intravenous immunoglobulin (IVIG) [23]. Over the subsequent 4 years, she had a gradual resolution of distal weakness, with some brief recurrences of weakness during the initial tapering of steroids and IVIG. By the 6th year after onset, the neurologic exam showed a good recovery of distal muscle strength and deep tendon reflexes, and fully normalized EMG/NCS parameters. Symptoms and signs of erythromelalgia persist 12 years after initial onset. A second patient presented with early-onset erythromelalgia, with severe itching as well as pain evoked by heat and improved by cold. Whole-exome sequencing confirmed a pathogenic *SCN9A* variant. She had NCS findings interpreted as providing “evidence for a very mild length-dependent sensory neuropathy.” No other patient had clinical signs or EMG/NCS findings consistent with a large-fiber peripheral neuropathy.

### 3.6. Medication History

More than 20 treatments were reported by the patients. The median number of medications tried was 4 (IQR = 1–7, range = 0–15). Twenty-eight (67%) patients tried more than one medication. Overall, patients trialed a wide range of medications with a varied response of a benefit, summarized in Figure 4.

Topical analgesics (lidocaine or other compound creams) were the most frequently tried medication (*n* = 26; 61.9%). Topical compound cream formulations reported included amitriptyline/ketamine (*n* = 4), amitriptyline/ketamine/clonidine/lidocaine (*n* = 1), and ketamine/gabapentin (*n* = 1). One patient did not specify the exact compound cream formulation. Other topicals used were aloe vera, Epsom salt/tea tree oil/peppermint oil soak, Cerave anti-itch lotion, Lotrimin cream, and Mentholatum ointment.

Fifteen (35.7%) patients reported an extensive list of other medications that were not specifically searched for in this study, with eight (19.0%) having tried more than one of the following: cyproheptadine, hydrocodone/acetaminophen, ketorolac, naproxen, tramadol, propranolol, amitriptyline, methadone, clonidine, naltrexone, IV morphine, oxycodone, olanzapine, ranitidine, sertraline, risperidone, omeprazole, doxepin, hydroxyzine, IV ketamine, chlorpromazine, amlodipine, diphenhydramine, hydrocodone/acetaminophen, Aspercreme, and hydromorphone. For most of these, it was not reported whether they provided a benefit. However, six patients reported a benefit from hydrocodone, ketorolac, naltrexone, IV morphine, clonidine, and Aspercreme. Two of these patients both reported clonidine as providing a benefit. Six patients reported alternative medications/therapies, which included acupuncture, Reiki, craniosacral therapy, and topical cannabidiol (CBD) and tetrahydrocannabinol (THC).

### 3.7. Family History

Six (14.3%) patients reported another family member with an erythromelalgia diagnosis and twenty-two (52.4%) patients had at least one family member with some type of chronic, inflammatory, vascular, or neuropathic condition.

### 3.8. Differences between Patients with and without Confirmed SCN9A Variant

Out of the 42 patients in our sample, only 3 (7.1%) had a confirmed *SCN9A* genetic variant. Given the small number of patients with *SCN9A* gene variants, no statistically significant differences were observed between patients with a *SCN9A* variant and those without a *SCN9A* variant regarding the age of onset, average pain score, and worst pain score (*p* > 0.05). The age of onset tended to be younger for SCN9A+ patients (*n* = 3, median = 4 years, range = 1–5 years) than SCN9A− patients (*n* = 37, median = 13 years, IQR = 3.75–14.25 years). No differences were observed for the worst pain scores reported by SCN9A+ patients (*n* = 3, median = 8, range = 7–10) compared to SCN9A− patients (*n* = 25, median = 8, IQR = 8–10). However, SCN9A+ patients tended to report higher average pain scores (*n* = 3, median = 7, range = 5–9) when compared to SCN9A− patients (*n* = 25, median = 5.25, IQR = 4.6–7).

## 4. Discussion

Here, we report on the clinical characteristics of 42 pediatric patients with erythromelalgia. We believe this cohort to be the largest pediatric erythromelalgia case series to date.

### 4.1. Clinical Characteristics

Similar to previous adult and pediatric studies [1,4,21], this cohort showed a female predominance, with a female to male ratio of 2.5:1. The age range of the onset of erythromelalgia in our cohort is similar to Arthur et al.’s systematic review [3], but the age of diagnosis of erythromelalgia was slightly younger than the retrospective review conducted by Cook-Norris et al. [4]. The younger age of diagnosis might possibly be attributed to the higher awareness of the diagnosis of erythromelalgia in the past 15 years compared to the era between 1970 and 2007.

Nevertheless, the age at diagnosis can be impacted by many factors, such as duration before seeing a specialist for symptoms, the number of specialists seen, types of specialists seen, and how familiar these specialists are with diagnosing erythromelalgia. Most patients in this cohort saw 2–3 specialists, with the most frequent being pain specialists, rheumatologists, and neurologists. Since our pain clinic was the primary referral center for those included in this study, this inevitably impacted the higher frequency with which this cohort saw pain specialists. In a 2021 survey (predominantly involving adults) conducted by The Erythromelalgia Association (TEA), the providers that made an erythromelalgia diagnosis most frequently were neurologists, dermatologists, and rheumatologists [24].

All four extremities were involved in 88% of patients. Among other affected body parts, the face and ears were the most frequent. One female patient reported pain in the groin area. These findings are consistent with other published studies and case reports reporting the involvement of the face, ears, and/or genitals [1,4,25]. Facial involvement has been reported as the third most common location of erythromelalgia symptoms after distal extremities, and the involvement of the genitals has been reported in about 2–3% of patients [1,5].

Symptoms were most frequently exacerbated by exercise/strenuous activity, heat, and ambient temperature changes. These modifying factors are consistent with those reported by a number of published studies [2,3,21]. Although cold is typically associated with relief in most erythromelalgia cases, six patients from this cohort reported cold worsening their symptoms. Three of these six patients (all sisters) specified extreme cold as worsening symptoms, despite being alleviated by milder cold exposure. In the other three cases, patients felt that their responses changed over periods of several years, with cold initially providing relief and later causing pain exacerbation.

An inciting event was reported by 36% of patients. The most commonly reported type of inciting event was infection. Among 175 primarily adult respondents in the Erythromelalgia Survey 2021 [24], the most commonly reported triggers were mental stress, surgery, and injury. Adult–pediatric comparisons in types of inciting events should be cautious, given the small numbers and difference between survey-based versus electronic health record-based study designs.

Burning and itching/tingling were the most common pain descriptors, though most patients reported multiple pain qualities and other sensory symptoms. In future studies, additional standardized questionnaires, such as the Adolescent Pediatric Pain Tool, may be beneficial in assessing multiple dimensions including sensory, affective, evaluative, and temporal descriptors of pain [26].

The swelling of feet and/or hands was reported for 17 patients (40%). Among these, seven had a history of abnormal CRP testing [7,14]. No patient in our current cohort was reported to have a myeloproliferative disorder. Of note, only one of the patients had secondary erythromelalgia associated with a confirmed inflammatory or metabolic diagnosis.

Since erythromelalgia presents with pain, warmth, and erythema in distal extremities, it is natural to consider possible roles of small-vessel vasculopathy in its pathogenesis. Non-invasive techniques for assessing micro-circulation [27] and large-vessel vasculopathy [28] are under evaluation for other pediatric diseases and could be considered in future prospective studies.

Depression was noted in 21% of patients in our sample. This frequency should be interpreted with caution, given the retrospective study design based on notes in electronic health records. Depressive symptoms and the diagnosis of major depressive disorder can co-occur with chronic pain, but reported frequencies vary widely among different case series [29]. National surveys of adolescents indicate an increasing prevalence of major depression over the past decade [30].

### 4.2. Treatment and Therapies

The most effective non-pharmacological strategies for relief reported by our cohort were ice pack/cooling gels, immersion in cold water, the use of air conditioning/fans, and resting/sitting.

Patients received a wide range of medications and frequently tried multiple medications, and no consistent pattern emerged regarding effectiveness. None of the patients followed at BCH were prescribed opioids on a chronic basis for erythromelalgia. Arthur and colleagues reported more than 40 medications tried in their case series, with variable and limited efficacy data [3].

### 4.3. Challenges in Gene Discovery

Only 6 of the 42 patients had a history of more than one affected family member and only 3 of the 42 patients had confirmed *SCN9A* variants. This confirms previous impressions of Cook-Norris et al. [4] that a majority of pediatric cases of erythromelalgia appear to be sporadic rather than inherited. The reported frequencies of sodium channelopathies and other inherited forms of erythromelalgia will also vary with the type of genetic testing performed. Whole-exome and whole-genome sequencing may identify variants not reported by clinical testing panels that involve the targeted sequencing of the *SCN9A* gene. As noted by Klein et al. [8], some previously reported *SCN9A* variants may be polymorphisms of an uncertain pathogenic significance or may have a modifier role rather than being causal per se.

Arthur et al. [3] compared features of pediatric erythromelalgia patients with and without *SCN9A* variants from among their own series of 13 patients at Great Ormond Street Hospital for Children and from 25 pediatric patients with *SCN9A* variants from their systematic review. In their series, patients with *SCN9A* variants had an earlier age of onset, a greater severity of pain, poorer responses to analgesic management, and more frequent skin complications compared to those without pathogenic *SCN9A* variants [3]. Among the different *SCN9A* variants, there were notable associations between channel electrophysiologic properties in vitro and clinical features, including age of onset and severity. In our current series, the age of onset of the three patients with confirmed *SCN9A* variants was younger (1, 4, and 5 years) than the median of the rest of the sample (13 years), but statistical comparisons are limited by the small numbers.

Differences in the percentage of erythromelalgia patients with confirmed gene variants in different series may reflect multiple factors, such as differing base frequencies in different populations, referral patterns, or different patterns of causation for acquired cases. As more novel gene variants are elucidated, future management may benefit from genetic analyses to better inform clinicians, patients, and families of the most effective treatment course.

Due to differences in the penetrance and expressivity of variants, there is a large variability in the genotypes and their associated phenotypes that manifest into rare pediatric disorders [31]. There is growing evidence that erythromelalgia involves various genetic variants in other genes besides *SCN9A*, many of which still remain unknown. Gene discovery to establish a database of known variants is limited by the availability of these types of patients, and the current existing research on pediatric erythromelalgia is confined to small sample sizes. There can also be discrepancies in merging clinically definite diagnoses and research-based diagnoses of erythromelalgia. Therefore, the Manton Center aims to counter these obstacles by developing a repository databank of genetic samples, from participants, that can be accessed by other researchers investigating orphan diseases, such as erythromelalgia. Together with the Manton Center, this research group has planned a multi-center collaboration to further supplement the existing information gathered from EHR.

In this patient cohort, besides the three patients with confirmed *SCN9A* variants, there were a number of gene variants of uncertain but possible pathogenic significance. However, these variants represent mostly novel gene candidates that are still under investigation for potential clinical and genetic significance and are therefore not reportable at this time.

### 4.4. Study Limitations

Electronic health records can be useful in collecting data on rare conditions and combatting small sample sizes of such studies. The availability of EHR data from multiple institutions can provide a number of patients with a particular rare disorder that is several-fold higher than that seen by just one provider or one hospital [32]. However, clinician notes in EHRs are primarily text fields with descriptive or narrative responses rather than structured inputs designed for research purposes. This necessitates pragmatic decision making to reach a unified method of data coding, data inclusion, and the visual representation of these findings. While outside medical records were included for a number of patients, there is the potential for missing data and loss to follow up.

Laboratory testing was at the discretion of the individual clinician. There was little uniformity in decisions regarding the inclusion or omission of specific tests. In the future, as part of a collaborative effort involving pain physicians, rheumatologists, neurologists, and dermatologists, we hope to generate recommendations regarding indications for the inclusion of specific laboratory tests.

Conclusions regarding the frequency of small- or large-fiber neuropathy in our patient cohort should be cautious, since only four patients underwent skin biopsy, no patient underwent sural nerve biopsy, and only ten patients underwent EMG/NCS studies [14,33,34]. The deferral of skin biopsy and limited use of EMG/NCS reflects prevailing practice among the pediatric neurologists and pediatric rheumatologists at BCH. In general, they defer the use of EMG/NCS in the absence of signs of large-fiber dysfunction on neurologic examination, such as weakness, hyporeflexia, or a loss of proprioception or vibration sense. Similarly, the pediatric neurologists and pediatric rheumatologists at BCH make limited use of skin biopsies, in part based on a reluctance to use results of the skin biopsy to support the treatment of small-fiber neuropathy with auto-immune- or auto-inflammatory-based therapies such as steroids or IVIG [35].

The role of small-fiber neuropathy in erythromelalgia as well as in several types of pediatric chronic pain is a point of controversy. Oaklander and colleagues have provided evidence for small-fiber involvement based on the use of neurite counting in small skin punch biopsies as a primary aid in diagnoses [15,36]. We share the hope of Oaklander and colleagues that a descriptive term like erythromelalgia can someday be replaced by disease definitions based on a mechanism. Conversely, we believe that it is premature, based on the current state of knowledge, to state with confidence that most or all cases of pediatric erythromelalgia are due to small-fiber neuropathy.

Even for those cases for which the clinical presentation, skin neurite counting, and functional tests such as thermoregulatory sweat testing, autonomic reflex screens, and quantitative sudomotor axon reflex testing are consistent with small-fiber neuropathy, the detailed causal role of auto-immune and auto-inflammatory mechanisms remains unclear. A recent randomized trial of intravenous gamma globulin for small-fiber neuropathy in adults failed to show a benefit [35].

Other factors that may impact the treatment of patients with chronic pain include health disparities, such as access to care and insurance, socioeconomic barriers, and racial and ethnic disparities [37]. In the current study, there was a wide variability in the number of clinician notes per patient for the management of erythromelalgia and a number of patients that were lost to follow up in the clinic. For prospective studies, these patients should be contacted, if amenable, to provide updated data regarding their current course of treatment.

## 5. Conclusions

Our single-center cohort of 42 pediatric patients with erythromelalgia showed variability in clinical characteristics. Many had identifiable triggering events, and only a minority of cases appeared to be inherited. No treatments emerged as consistently effective. We believe this cohort to be the largest pediatric study of erythromelalgia to date. Further research should investigate underlying mechanisms, subtypes of erythromelalgia, and responses to treatment. Planning is in progress for a pediatric multi-center registry and prospective survey study.

## Figures and Tables

**Figure 1 children-10-01282-f001:**
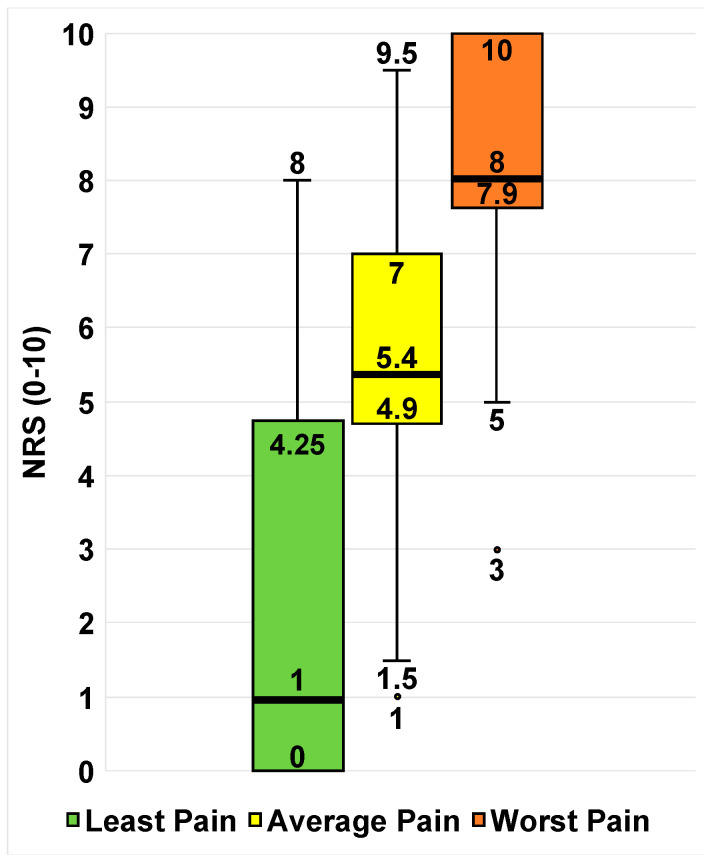
Available least, average, and worst pain scores of the sample (*n* = 28). Box-and-whisker plots of self-reported pain scores for least pain (green), average pain (yellow), and worst pain (red). The ends of the whiskers represent the range values. The horizontal line across the middle of the box represents the median. The bottom and top borders of the boxes represent the 25th and 75th percentiles, respectively. Any points outside of the range denote an outlier in that category.

**Figure 2 children-10-01282-f002:**
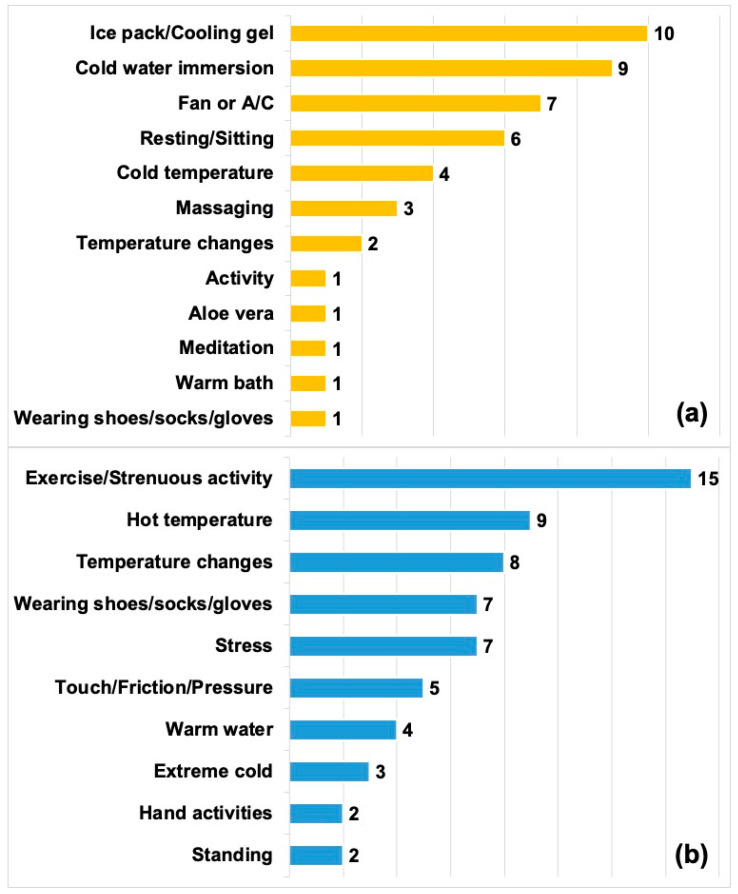
Modifying factors. (**a**) Factors that improved symptoms (*n* = 23). (**b**) Factors that exacerbated symptoms (*n* = 32). Bar values add up to more than 23 and 32, respectively, as several patients reported more than one modifying factor.

**Figure 3 children-10-01282-f003:**
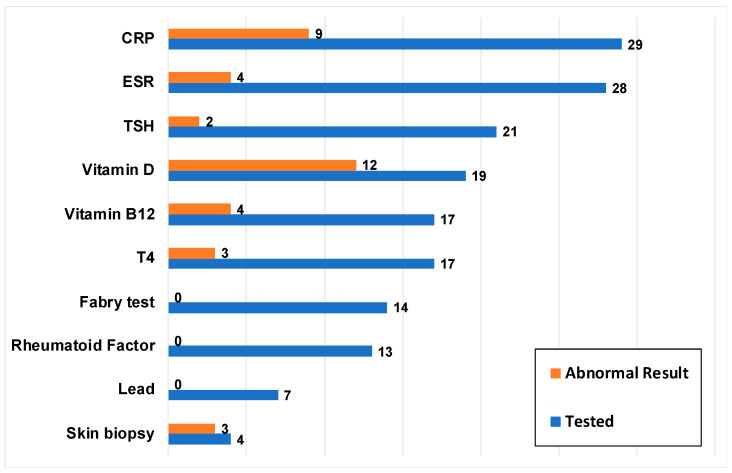
Laboratory tests performed (*n* = 38). Blue bars represent the number of patients who underwent a particular laboratory test and orange bars represent how many of these patients had an abnormal test result at any time point. C-reactive protein (CRP); Complete blood count (CBC); Erythrocyte sedimentation rate (ESR); Thyroid-stimulating hormone (TSH); Thyroxine (T4).

**Figure 4 children-10-01282-f004:**
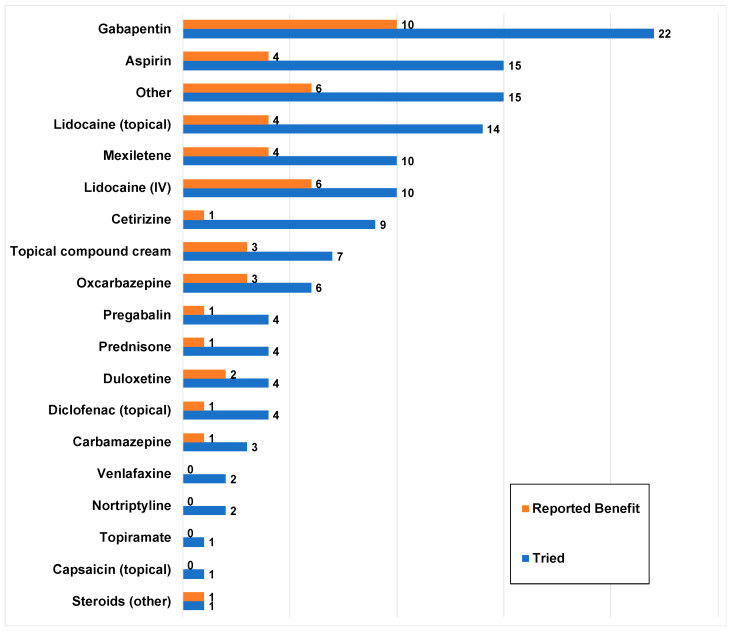
Medications tried by the sample and their perceived benefit (*n* = 42). Blue bars represent the number of patients who tried a particular medication and orange bars represent how many of these patients reported a perceived benefit from the medication.

**Table 1 children-10-01282-t001:** Demographics, inciting event, and other comorbidities (*n* = 42).

Data Type	N	%
Sex		
Female	30	71.4
Male	12	28.6
Inciting Event		
Infection	9	21.4
Physical activity	3	7.1
Trauma	3	7.1
Not reported	27	64.3
Other Diagnosis		
Anxiety	12	28.5
Migraines	11	26.2
Abdominal pain ^1^	10	23.8
ADHD ^2^	9	24.4
Depression	9	21.4
Autism	6	14.3
Dysmenorrhea/Endometriosis	6	14.3
Joint pain ^3^	6	14.3
Developmental delay	5	11.9
Small-fiber neuropathy	4	9.5
Learning disability	3	7.1
Seizures	2	4.8
CRPS ^4^	1	2.4
CIDP ^5^	1	2.4
POTS ^6^	1	2.4
PTSD ^7^	1	2.4
Occipital neuralgia	1	2.4
Trunk pain	1	2.4

^1^ Abdominal pain includes irritable bowel syndrome; ^2^ Attention-deficit/hyperactivity disorder; ^3^ Joint pain includes juvenile idiopathic arthritis; ^4^ Complex regional pain syndrome; ^5^ Chronic inflammatory demyelinating polyneuropathy; ^6^ Postural orthostatic tachycardia syndrome; ^7^ Post-traumatic stress disorder.

**Table 2 children-10-01282-t002:** Specialists consulted (*n* = 42).

Type of Specialist	N	%
Pain specialist	30	71.4
Rheumatologist	23	54.8
Neurologist	22	52.4
Dermatologist	12	28.6
Geneticist	10	23.8
Immunologist	2	4.8
Allergist	2	4.8
Vascular anomalies	2	4.8
Orthopedic	1	2.4

Pain specialists include those in the Pain Clinic at Boston Children’s Hospital and in outside clinics.

**Table 3 children-10-01282-t003:** Pain and clinical characteristics.

Characteristic		N	%
Affected body part (*n* = 42)	Bilateral feet	38	90.5
	Bilateral hands	37	88.1
	Face	13	31.0
	Ears	10	23.8
	Joints	8	19.0
	Legs	4	9.5
	Neck	2	4.8
	Whole body	2	4.8
	Arms	1	2.4
	Groin	1	2.4
Pain quality (*n* = 36)	Burning	28	66.7
	Itching/Tingling	16	38.1
	Paresthesia/Pins and needles	8	19.0
	Shooting/Stabbing/Sharp	7	16.7
	Achy/Throbbing	6	14.3
	Numbness	4	9.5
Extremity coloration (*n* = 38)	Red only *	27	64.3
	Multicolor **	9	21.4
	Pink only	1	2.4
	Purple only	1	2.4

* ‘Only’ signifies that this was the only color reported in the clinical notes and not necessarily that no other color changes ever occurred. ** Multicolor includes various combinations: red/purple, red/mottled, red/purple/blue/white/orange, red/purple/pink/pale/mottled, and red/blue/black/white.

## Data Availability

De-identified data presented in this study are available upon request from the corresponding author.

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
