# Peer review of "Clinical Characterization of Pediatric Erythromelalgia: A Single-Center Case Series"

_children, 2023, doi:10.3390/children10081282_

Round 1

Reviewer 1 Report

I have reviewed the abstract, introduction, methods and materials, results, and conclusion. I have also checked the references, and all appear relatively current and appropriate. Finally, I have also reviewed the figures, tables, and legends.

I find the case series well-written, well-done, and informative.

However, I have one minor issue:

1. The inclusion and exclusion cryteria are a little bit vague. Could You explain?

Reviewer 2 Report

There are many line breaks in the Introduction and Discussion, so please adjust them accordingly.

At what time are the data in Table 1, Table 3, and Figure 3? At diagnosis? Last observation? Other?

Please discuss the prevalence of depression in Table 1.

Are pediatricians excluded from Table 2?

At what time are the data in Figures 2 and 4? Last observation? Other?

Does Figure 4 exclude medications used for conditions other than erythromelalgia, such as complications?

Please add the treatment details for the 6 patients who were beneficial in Other in Figure 4. 

Reviewer 3 Report

Sun et al provide a single - center cross- sectional study evaluating clinical characteristics, associated conditions and treatment response  of pediatric patients with erythromelalgia. 

I have only some minor comments: 

- please present more characteristics of the patients, for e.g. laboratory parameters such as creatinine, c- reactive protein for e.g. in a table 

- only 3 patients had confirmed SCN9A variants - are there hypotheses about the disease etiologies of the other patients? For e.g. after viral infection?

- did you perform microcirculatory measurements? If so, please include into the results. 

- moreover, further vascular measurements, for e.g. IMT in carotid ultrasound would be of benefit for the follow u, to asses cardiovascular risk. 
